# Revealing Angiopep-2/LRP1 Molecular Interaction for Optimal Delivery to Glioblastoma (GBM)

**DOI:** 10.3390/molecules27196696

**Published:** 2022-10-08

**Authors:** Angela Costagliola di Polidoro, Andrea Cafarchio, Donatella Vecchione, Paola Donato, Francesco De Nola, Enza Torino

**Affiliations:** 1Department of Chemical, Materials and Production Engineering (DICMaPI), University of Naples Federico II, 80125 Naples, Italy; 2Interdisciplinary Research Center on Biomaterials, CRIB, University of Naples Federico II, 80125 Naples, Italy; 3Dipartimento di Medicina e Scienze della Salute Vincenzo Tiberio, University of Molise, 86100 Campobasso, Italy; 4Teoresi Spa—Via F. Imparato, 198 CM2, 80146 Napoli, Italy

**Keywords:** angiopep-2, glioblastoma, optimal design, molecular docking

## Abstract

Background: The family of synthetic peptide angiopeps, and particularly angiopep-2 (ANG-2) demonstrated the ability preclinically and clinically to shuttle active molecules across the blood–brain barrier (BBB) and selectively toward brain tumor cells. The literature has also proved that the transport occurs through a specific receptor-mediated transcytosis of the peptide by LRP-1 receptors present both on BBB and tumor cell membranes. However, contradictory results about exploiting this promising mechanism to engineer complex delivery systems, such as nanoparticles, are being obtained. Methodology: For this reason, we applied a molecular docking (MD)-based strategy to investigate the molecular interaction of ANG-2 and the LRP-1 ligand-binding moieties (CR56 and CR17), clarifying the impact of peptide conjugation on its transport mechanism. Results: MD results proved that ANG-2/LRP-1 binding involves the majority of ANG-2 residues, is characterized by high binding energies, and that it is site-specific for CR56 where the binding to 929ASP recalls a transcytosis mechanism, resembling the binding of the receptor to the receptor-associated protein. On the other hand, ANG-2 binding to CR17 is less site-specific but, as proved for apolipoprotein internalization in physiological conditions, it involves the ANG-2 lysin residue. Conclusions: Overall, our results proved that ANG-2 energetic interaction with the LRP-1 receptor is not hindered if specific residues of the peptide are chemically crosslinked to simple or complex engineered delivery systems.

## 1. Introduction

Effective delivery to the central nervous system (CNS) is an incredibly challenging task that is still mostly unaccomplished despite the numerous efforts outlined in the recent literature on this topic. Invasive and non-invasive approaches, both relying on externally applied stimuli or physiological mechanisms, as well as different routes of administration have been extensively explored to improve active agent penetration into the CNS [1,2].

The angiopep family of peptides has been derived from the Kunitz domain of human aprotinin [3]. These peptides are able to cross the blood–brain barrier (BBB) and have been used to facilitate the delivery of pharmacological agents to the brain, for example to target glioblastoma tumors and recurrent brain metastases of pre-treated breast cancers [4]. In particular, angiopep-2 (ANG-2) has higher transcytosis capacity and higher brain volume of distribution than aprotinin. Like aprotinin, angiopep-2 interacts with low-density lipoprotein receptor-related protein 1 (LRP1) which is thought to promote its delivery across the BBB via receptor-mediated transcytosis (RMT) [5]. The LRP-1 receptor is a 600 kDa member of the low-density lipoprotein receptor family physiologically involved in the transcytosis of many proteins and peptides across the BBB [6]. 

Both natural and synthetic ligands have been explored for the targeting of LRP-1 receptors, and among them, the synthetic peptide angiopep-2 has gained increasing interest over the years [4,5,7,8,9,10,11,12,13]. From its synthesis in 2008, the 19-amino-acid peptide ANG-2 has demonstrated huge transcytosis potential on endothelial cells of the BBB based on its selective interaction with LRP-1 receptors, confirmed both in preclinical and clinical settings [14]. The formulation ANG1005 [13], a drug–peptide conjugate in which a molecule of ANG-2 is conjugated to three paclitaxel (PTX) molecules, improved drug BBB penetration and brain parenchyma accumulation up to 10-fold compared to the free drug in rats [13] and it is currently in a clinical trial for human use [15]. In 2015, Tian et al. measured the ability of ANG-2-functionalized polymersomes to cross a transwell-based model of the BBB revealing that ANG-2-functionalized NPs were shuttled across the bEnd.3 cells more effectively than all other non-functionalized formulations [16]. Additionally, Xin et al. [8], proved that the conjugation of ANG-2 to the surface of PEG-PCL NPs leads to LRP-1-mediated improved transport at the endothelial layer of the BBB in vitro but also to an improved accumulation in the brain parenchyma of healthy and tumor-bearing mice in vivo, proving first that the targeting ability of the peptide is independent of the presence of disrupted BBB associated to the late stages of the tumor, and second, its ability to target tumor cells. Thus, starting from the evidence that ANG-2 bears dual targeting ability, being additionally able to improve selective uptake by glioma cells, we designed ANG-2-engineered theranostic hydrogel nanoparticles (Thera-ANG-cHANPs) for improved therapy and boosted imaging of glioblastoma multiforme [12]. We proved that Thera-ANG-cHANPs uptake in glioma patient-derived cells is significantly boosted with respect to bare particles and that, as a consequence of the improved transport into cells, engineered nanoparticles enhance the therapeutic efficacy of the delivered chemotherapeutic drug irinotecan.

Despite this widespread success, the literature still presents some contradictory evidence regarding the active targeting potential of angiopep-2-engineered nanoformulations for brain tumor delivery, and the interaction with LRP1 may not be the only method for angiopep-2 to cross into the brain. Indeed, although extensive work has been done in determining the LRP-1 involvement in ANG-2 transcytosis [7,10,13], the characterization of ligand/receptor energetic interaction as well as the identification of both receptor binding sites and ANG-2 targeting moieties are still missing. This knowledge can clarify the impact that the peptide conjugation to particle surface might have on the targeting ability of the peptide, and make it possible to hypothesize on the intracellular trafficking of the ligand/receptor complex, which might impact its overall in vivo effectiveness. 

In this framework, molecular docking is a powerful computational tool widely applied to studying and characterizing the energetic interaction of small molecules with the binding site of target proteins at the molecular level [17]. Docking methodologies support the prediction of the experimental binding modes and affinities of selected ligand/receptor pairs and are currently used as a standard computational tool in drug design [18]. Over the years, this methodology has found extensive application in the virtual screening of drugs, selection of new biologically active molecules, and drug repurposing [17,18,19]. In addition to its traditional implementation, molecular docking has appealing characteristics for the precision medicine field and particularly for the optimal design of targeted delivery of nanoparticles. 

Indeed, by revealing and characterizing the ligand/receptor interaction at the molecular level, molecular docking can aid in the rational design of engineered drug delivery systems, the revelation of ligand targeting sites, and consequently the determination of the most appropriate conjugation methodology that, involving specific ligand residues, might impact the ligand targeting ability.

In the present study, we propose a molecular docking-based tool to characterize for the first time the energetic interaction of the ANG-2/LRP1 complex, finding a molecular basis for the success that this peptide has demonstrated in clinics. Flexible docking is applied to identify ANG-2 binding sites on the LRP1 receptor and particularly on its complement-type repeats (CRs) 56 and CR 17 which are reported as the major ligand-interacting receptor sites in physiological conditions [18,20]. In detail, the double-module CR56 is the portion of LRP-1 responsible for the specific interaction with many ligands such as the receptor-associated protein (RAP) which has been proved to be involved in the process of receptor recycling to the cell surface, and thus directly implicated in its transcytosis properties [21]. Moreover, CR17 is the portion of LRP-1 responsible for lipoprotein internalization, particularly ApoE. 

For these reasons, this methodology offers an improved understating of the ligand–receptor complex formation and determines suitable ANG-2 moieties at binding, allowing the exploitation of the targeting potential of this peptide. 

## 2. Materials and Methods

### 2.1. Angiopep-2 3D Structure Prediction

Angiopep-2 is a 19-amino-acid synthetic peptide (TFFYGGSRGKRNNFKTEEY) that has been designed to specifically interact with LRP-1 receptors [3], which are of great interest for the crossing of the blood–brain barrier and glioma cell targeting.

As a very novel peptide, the angiopep-2 (ANG-2) 3D model is not available in standard databases. For this reason, to perform the molecular docking simulation and study its interaction with the LRP-1 receptors, its three-dimensional structure is obtained by the software PEP-FOLD3. PEP-FOLD3 allows the determination of the 3D structure of linear peptides of 5–50 amino acids in an aqueous solution. The folding process proceeds by adding one amino acid at a time along the whole amino acid sequence and assures very high reliability. Indeed, Lamiable et al. [22] proved that PEP-FOLD 3 prediction of peptide folding occurs with a deviation of only 3.3Å from the experimental conformation.

Starting from the given amino acid sequence, PEPFOLD3 generates 100 models that are clustered using Apollo [23] to identify similar models. The clusters are then sorted using either the sOPEP energy value or Apollo-predicted TM-score (tm). As a result of this clustering, five out of the most probable conformations are extracted and the probability of folding structures (e.g., alpha-helix, β-sheets) in a 2D plot is presented in Figure 1. 

### 2.2. Relevance of LRP-1 CR56 and CR17 in Ligand-Binding

It has been reported that different local charge distributions are of fundamental importance for the specificity of the binding of ligands to specific complement repeats (CRs) in LRP-1 [20]. Indeed, despite their similarity due to the short amino acid sequence (about 40 amino acids per CR) and their open-loop structure, each CR guarantees specificity against ligands mainly by different contour surfaces and charge distributions [20].

The double module of complement-type repeat CR56 is the portion of LRP-1 in the cluster II binding site responsible for the specific interaction with many ligands such as the receptor-associated protein (RAP). The RAP is an endoplasmic reticulum (ER)-resident protein required for the effective recycling of LRP to the cell surface, and for this reason, it is involved in the transcytosis process characterizing the transport of many substrates of the receptor. Indeed, it prevents premature interaction between the receptor and ligands in the ER, thereby preventing the receptor from being degraded in lysosomes [21]. LRP-1–RAP binding has been extensively studied and CR56 of complex II has been identified as the main site of the binding [20].

CR17 in cluster III is reported as the primary binding site for apolipoproteins and specifically for ApoE [24]. Lipid association of ApoE can enhance receptor binding by several mechanisms. Since multiple copies of ApoE are embedded in lipoprotein particles, strings of CRs could bind to several ApoEs at once, creating an avidity effect [25]. Guttman et al. [24] demonstrated that the ApoE helix contacts CR17 on the side rather than directly at the calcium-binding site.

The flexible docking of the novel peptide angiopep-2 with both CR56 (PDB ID: 2FYL) and CR17 (PDB ID: 2KNX) of LRP-1 is performed to energetically characterize the ANG-LRP1 binding, identify angiopep-2 residues involved in the binding and understand whether the chemical conjugation of this peptide to active molecules or nanocarriers might interfere with its targeting ability. The PDB files of both CR56 and CR17 were downloaded from Protein Data Bank website [26,27] as .pdb files and directly opened in Autodock Vina. Before running the simulation, both receptors and ligands were prepared according to the steps described in the next section.

### 2.3. Flexible Docking by Autodock Vina

The ligand–receptor interaction was long taught as a lock and key mechanism in which the ligand fits rigidly in the receptor binding site, according to the theory proposed by Fischer [28]. This theory, however, has more recently given way to the “induced fit” theory highlighting the importance that continuous protein reshaping has in ligand–receptor binding. According to this theory, both the ligand and the receptor should be considered flexible despite a trade-off between accuracy and computational time that can be achieved by considering the ligand flexible and the receptor rigid [17]. 

In the present study, we perform a flexible docking simulation by keeping the receptor rigid and the ligand flexible. Indeed, the compromise of keeping the receptor fixed allowed us to perform flexible docking simulations with high accuracy and reduced simulation time with the tool AutoDock. AutoDock is a suite of automated docking tools designed to predict how small molecules, such as substrates or drug candidates, bind to a receptor of a known 3D structure. Autodock Vina achieves significant improvements in the average accuracy of the binding mode predictions while being up to two orders of magnitude faster than AutoDock4, and for this reason it is chosen for the simulations [29]. As a result of docking, the energy of the binding in kcal/mol is obtained. To gain information about the nature of the binding as well as the residues involved, and thus to identify the angiopep-2 binding site, the open-source software PyMOL and, in particular, OpenGL Extension Wrangler Library (GLEW) and FreeGLUT plugin able to solve Poisson–Boltzmann equations using the Adaptive Poisson Boltzmann Solver, are used. The operating procedures are set as follows: (1) load the molecule in .pdb format; (2) select the area in which the software will compute the calculation and obtain its coordinates with the tool Gridbox. The Grid Center is set at (55.048; 1.86; −2.473) as (x; y; z) and a spacing of 0.375 Å is chosen; (3) save the macromolecule in .pdbqt format; (4) list receptor and ligand filename and insert the selected area coordinates; (5) select 10 as the number of tries; (6) choose the exhaustiveness (accuracy of the simulation). To analyze the obtained results, for each simulation: (1) open the result file with block notes to obtain the binding affinity value; (2) load the result file in Pymol; (3) change the visualization of the molecules in stick and balls; (4) select the function show residues; (5) select the legacy plug-in show contact; (6) select the atoms involved in the binding and export the list of the involved residues as an .xls file. The residues involved in the binding are selected as statistically relevant or not. This selection is performed by building a Gaussian distribution for each simulation representing the probability of occurrence of each binding among different trials. All the binds with a probability of occurrence included in three times the standard deviation of the Gaussian distribution are selected as statistically relevant (please note that this procedure led to selection as statistically relevant in almost 98% of the residues identified by Autodock).

## 3. Results

### 3.1. Protein–Protein Flexible Docking: Angiopep-2 Binding to CR56

LRP-1 is a 600 kDa protein overexpressed at the endothelial layer of the BBB and involved in the physiological transport of lipoprotein inside the central nervous system. It is composed of four different clusters, with clusters II and III being mainly responsible for ligand binding.

The double-module, complement-type repeat CR56 constitutes the ligand-binding site of cluster II and is composed of 82 residues, from 932Ser to 1013His [20]. In particular, CR5 involves the acidic residue 959Asp, which does not coordinate calcium. This residue has been proven to be involved in the binding to the receptor-associated protein (RAP), and for this reason it is considered an active player in selective ligand binding and transcytosis [20]. Further details about the mechanism of transcytosis mediated by RAP binding are presented in the dedicated methods section.

The CR56 3D model was used to perform flexible docking simulations with all five ANG-2 models predicted by PEP-FOLD3, and the results were compared in terms of the binding site, binding energy, and nature of the binding. Figure 2 visually reports the results of the simulation showing similar regions of LRP-1 involved in the binding for each of the ANG-2 models. The binding energies and LRP-1 residues involved are reported in Table 1. The results show that all the models interact very similarly with CR56, with binding energies characterized by a mean of −5.68 ± 0.6 kcal/mol except for model 2, showing higher binding energy that, however, correlates with a higher standard deviation (7.8 ± 1.2 kcal/mol).

Moreover, the docking results showed that each ANG-2 model binds 959Asp of LRP-1 (29Asp of CR5) and that the binding occurs through electrostatic clashes that characterize 29% of the total binds. This result confirms that ANG-2 binding to CR56 of LRP-1 mimics the interaction with the RAP protein, possibly triggering a transcytosis mechanism [5] and that this selective binding occurs through electrostatic interactions, as previously reported for other ligands [20].

In the second step, interacting ANG-2 residues were analyzed and the results showed that almost all ANG-2 residues bind LRP1 (Figure 3). The plots present the frequency of residues involved in the binding as it occurs in the 10 simulation repeats. The involvement of all the residues is not a surprising result, considering how this synthetic peptide has been designed. Demeule et al. [3] obtained the amino acid sequence of ANG-2 by comparing the sequences of physiological substrates of the receptor and composing the common moieties differently, comparing the transcytosis capacities of the resulting peptides. ANG-2 emerged as the optimized sequence with the highest transcytosis capacity on a model of the BBB, confirming its potential preclinically and clinically [13]. Despite the fact that, in some cases, such as for ANG-2 models 1 and 3, a few residues are mainly responsible for the binding, in models 2, 4, and 5 almost all the residues appear to be involved. These results allow concluding that each ANG-2 residue is involved with an almost equal probability in the binding of the CR56. This hypothesis turns even more probable considering that during the delivery process, from administration to tumor cell targeting, ANG-2 folding might dynamically change, according to all the possible five conformations, depending on the environmental conditions. As an example, the tumor microenvironment is characterized by a strongly reduced pH with respect to the physiological one, sometimes dropping down to 5.5, and this certainly impacts the folding of the peptide.

The main consequence of this result, showing all ANG-2 residues being involved in CR56 binding, is that the conjugation of this peptide with a drug delivery system that requires its chemical binding—thus preventing some residues from interaction with the receptor—does not negatively impact on the overall peptide targeting ability. Overall, molecular docking results allowed us to define the molecular pattern at the basis of the ANG-2/LRP-1 interaction, responsible for the transcytosis at the endothelial layer of the BBB and boosted uptake by glioma cells, as already proved by different authors [5,10,11,12,13].

### 3.2. Protein–Protein Flexible Docking: Angiopep-2 Binding to CR17

CR17 is the complement repeat of LRP1 cluster III involved in lipoprotein internalization. Regarding the transport of lipoprotein, LRP1 shares relevant interacting moieties with other receptors of the LDL family, such as LRP3 [24]. The common subdomains CR16–18 show particularly high affinity toward ApoE, with the highest affinity for the single repeat CR17. 

CR17–ANG-2 flexible docking was performed for all five models of ANG-2. As for CR56, Figure 4 visually reports the results of the simulation that, in this case, show similar but not identical regions of LRP1 involved in the binding for each of the ANG-2 models. Binding energies and LRP-1 residues involved are reported in Table 2. The results show a slightly reduced binding energy of ANG-2 to CR17 with respect to CR56, the average energy being equal to −5.3 ± 0.62 kcal/mol. Moreover, in this case, all models show similar binding energy, except for model 2 displaying a higher average value and standard deviation (−8 ± 1.46).

Upon the analysis of the nature of the binding it can be observed that, despite very similar binding energies, the relative amount of pi interaction and electrostatic clashes is significantly different between different models. Additionally, it was not possible to identify a specific binding pattern or a specific residue sequence involved in the binding, and for this reason it appears that CR17–ANG-2 interaction is less site-specific than CR56–ANG-2. Examining the results obtained for ANG-2 residues involved in the interaction with CR17, Figure 5 reveals that in almost all models, the majority of the amino acids are involved in the binding and that, as before, models 1 and 3 display the preferential interaction with specific amino acids. Additionally, it appears that all the residues are involved with a high frequency, meaning that they are almost simultaneously involved. This result might correlate with the avidity property of CR17 for apolipoprotein, referring to the possibility of multiple binding sites sometimes leading to simultaneous internalization of multiple ligands.

Additionally, the literature proposes a comprehensive characterization of ApoE binding to CR17, revealing that it interacts with LRP1 through lysine residues that bind the receptor in regions surrounding the calcium ion binding site identified between 25Trp and 26Leu of CR17 [24]. Neither 25Trp nor 26Leu characterizes the CR17–ANG-2 binding site, except for model 5, which binds 26Leu through both pi and electrostatic interactions. Despite neither 25Trp nor 26Leu being involved, ANG-2 presents two lysine residues in its sequence that are always involved in the binding, even in models 1 and 3, as in the case of ApoE binding. Overall, these results allow hypothesizing that CR17–ANG-2 binding mimics the CR17–ApoE binding closely both in terms of the nature of the binding and binding site. As a consequence, since ApoE internalization is crucial for cancer cell development [30], it might be speculated that this interaction could be the basis of nanoparticle-boosted uptake in tumor cells. Indeed, LRP-1 being abundantly spread over the cell membrane of tumor cells, and given the hypothesized avidity effect of ANG-2 binding to the receptor, this mechanism could be the basis of the increased uptake that has been observed in the literature both on standard cell lines and on patient-derived cells, as in [10] and the study by Costagliola et al. [12].

## 4. Discussion

Angiopep-2 (ANG-2) is a novel peptide that has demonstrated tremendous potential in improving the transport of active agents in the central nervous system and brain tumor targeting, crossing the BBB, and guaranteeing selective accumulation in glioma cells through specific interaction with LRP-1 receptors [3]. Despite its initial clinical success, the combination of this peptide with nanoparticles has led to promising but sometimes contradictory results [31]. We recently proved that ANG-2 conjugation to hydrogel nanoparticles boosts NP uptake by glioma patient-derived cells and improves the therapeutic effect of the delivered drug irinotecan, in accordance with the previous findings on ANG-2 dual targeting ability [12]. This effect, however, was not confirmed by other authors proposing the conjugation of ANG-2 on the surface of liposomes, obtaining no improvements either in BBB crossing or in tumor cell targeting both in vitro and in vivo [31]. 

In this work, we are proposing digital tools to understand the controversial results published about ANG-2, clarifying the role of this peptide in binding with LRP1, to support its clinical success and favor its rational use in the field of nanomedicine. Indeed, through a methodology based on molecular docking, we deepened the knowledge about the molecular interaction of ANG-2 with the targeting receptor LRP-1 to gain an improved understanding of the impact that peptide surface conjugation has on its targeting ability and intracellular trafficking. 

Our results demonstrated that ANG-2 binding to both CR56 and CR17, the primary receptor binding site for its physiological ligands, is characterized by high binding energy. First, ANG-2/CR56 binding involves the residue 929ASP of LRP-1 responsible for the receptor recycling to the cell surface, and thus the transcytosis process through the interaction with the RAP protein. Secondly, ANG-2 docking to CR17 revealed multiple and equally probable binding sites of ANG-2 on the complement repeat, closely resembling the avidity effect that characterized the ApoE internalization by glioma cells. Additionally, both the docking procedures revealed the involvement of all the ANG-2 residues in the interaction with the receptors, displaying an equal probability of occurrence. These results make it possible to conclude that ANG-2 affinity to the receptor LRP-1 is not significantly decreased if hindering specific ligand residues because of surface conjugation, following our previous findings. Moreover, the similarity with ApoE internalization and the determinant role of ApoE in guaranteeing tumor cell proliferation, angiogenesis, and metastasis, suggested that this internalization route might be preferential in the uptake by target glioma cells. 

In light of the obtained results, we can hypothesize that the failure of ANG-2 in providing boosted BBB crossing and tumor cell uptake of specific nanoformulations can be ascribed to the poor understanding of the connection between the synthetic identity of that specific nanovector and its biological identity more than an intrinsic ANG-2/LRP1 interaction failure. Indeed, the formation of a protein corona in serum, the loss of colloidal stability, and aggregation might prevent ANG-2 access to the receptor, hindering the binding. Similarly, specific synthetic identities might trigger multiple uptake mechanisms (e.g., membrane fusion, receptor-mediated endocytosis through different ligands), which might compete with ANG-2 RMT by LRP-1, lowering its targeting potential. 

For this reason, a comprehensive characterization of the connection between the synthetic identity of the ANG-2-engineered nanovectors and the biological identity they will acquire upon administration is the most reliable way for the rational use of the peptide, allowing its targeting potential to be exploited fully.

## 5. Conclusions

In conclusion, in this work, we clarified the molecular interaction of the synthetic peptide angiopep-2 with its target receptor LRP-1 through molecular docking. This knowledge will contribute to increasing the awareness around this peptide’s use as a targeting moiety in precision medicine. Moreover, it will provide a rationale for its conjugation to nanoparticle surface when optimally designed for glioblastoma targeting.

## Figures and Tables

**Figure 1 molecules-27-06696-f001:**
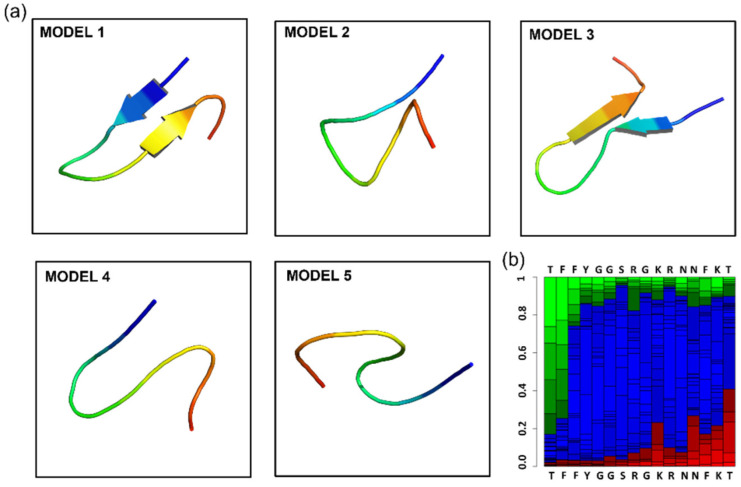
Prediction of angiopep-2 3D folding by PEP-FOLD3. (**a**) Five most probable 3D conformations of Angiopep-2; (**b**) 2D plot of folding probability in alpha-helix (green), β-sheet (red), or others (blue).

**Figure 2 molecules-27-06696-f002:**
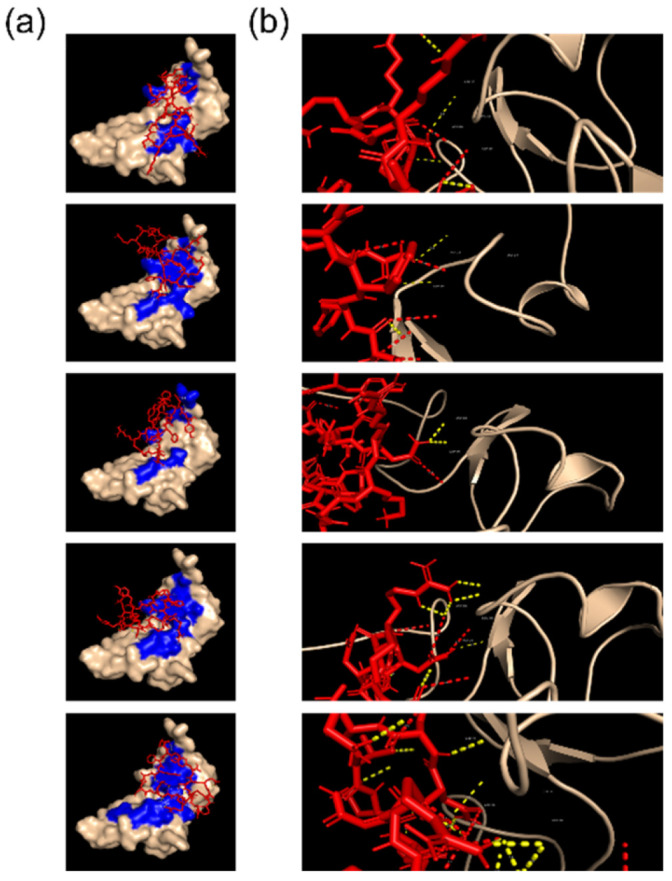
Simulation of Angioepep-2/CR56 binding: molecular docking results; (**a**) 3D representation of angiopep-2 binding mode on CR56 (in blue); (**b**) details of Angiopep-2/CR56 binding (dotted line in yellow represents pi interaction, in red electrostatic clashes).

**Figure 3 molecules-27-06696-f003:**
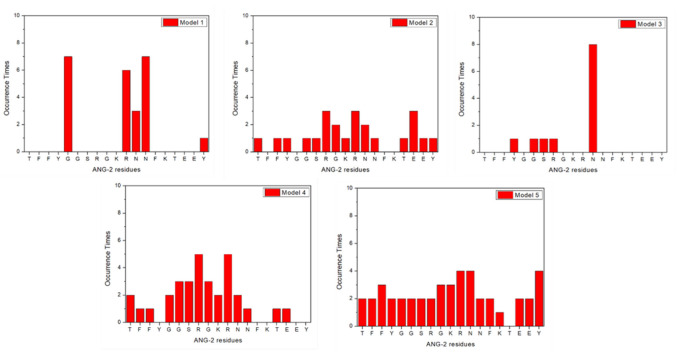
Angiopep-2 residues involved in LRP-1 binding from model 1 to model 5.

**Figure 4 molecules-27-06696-f004:**
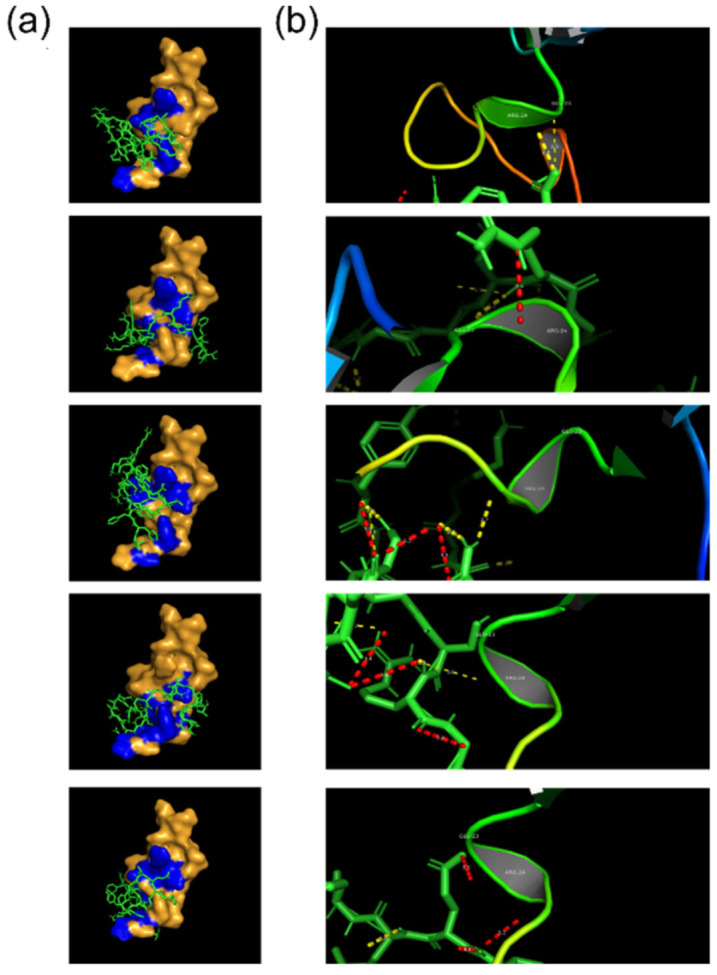
Simulation of Angioepep-2/CR17 binding: molecular docking results; (**a**) 3D representation of angiopep-2 binding site on CR17 (in blue); (**b**) details of Angiopep-2/CR17 binding (dotted line in yellow represents pi interaction, in red electrostatic clashes).

**Figure 5 molecules-27-06696-f005:**
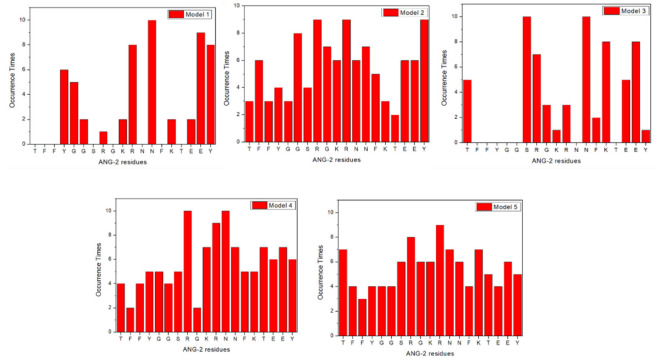
Angiopep-2 residues involved in LRP-1 binding from model 1 to model 5.

**Table 1 molecules-27-06696-t001:** Binding of angiopep-2 to CR56 of LRP-1: characteristic binding energy and classification of receptor residues involved in the binding.

	Binding Energy ΔG [=] Kcal/mol	Pi Interaction	Electrostatic Clashes
Model 1	−5.25 ± 0.2	16ARG, 27ASP, 28ASP, 30CYS, 62ANS, 70ASP, 80CYS	28ASP, 29ASP, 70ASP
Model 2	−7.8 ± 1.2	16ARG, 27ASP, 28ASP, 62ANS, 63TRP, 64ARG, 70ASP, 71CYS, 80CYS	29ASP, 62ANS, 70ASP, 80CYS
Model 3	−5.12 ± 0.2	16ARG, 28ASP, 29ASP, 62ANS, 65CYS, 80CYS, 82HIS	16ARG, 29ASP, 62ANS, 65CYS, 80CYS, 82HIS
Model 4	−5.22 ± 0.4	16ARG, 28ASP, 29ASP, 30CYS, 58CYS, 62ANS, 63TRP,70ASP, 71CYS, 80CYS	28ASP, 29ASP, 58CYS, 62ANS,70ASP, 71CYS, 80CYS
Model 5	−5.05 ± 0.3	16ARG, 17CYS, 25ASP, 27ASP, 28ASP, 29ASP, 30CYS, 58CYS, 63TRP,70ASP, 71CYS	17CYS, 27ASP, 28ASP, 29ASP, 70ASP, 71CYS, 80CYS

**Table 2 molecules-27-06696-t002:** Binding of angiopep-2 to CR17 of LRP-1: characteristic binding energy and classification of receptor residues involved in the binding.

	Binding Energy ΔG [=] Kcal/mol	Pi Interaction	Electrostatic Clashes
Model 1	−4.77 ± 0.2	17THR, 41ILE, 50THR, 24ARG, 47TYR,27CYR, 28ASP	17THR, 41ILE, 50THR, 27CYS, 28ASP
Model 2	−8 ± 1.4	11SER, 13SER, 27CYS, 39GLU, 41ILE, 48ASN, 43ALA, 23GLU, 24ARG	11SER, 13SER, 27CYS, 39GLU, 41ILE, 48ASN, 43ALA, 23GLU, 24ARG
Model 3	−5.05 ± 0.17	24ARG, 28ASP, 29GLY, 30ASP, 49SER, 23GLU, 47TYR	28ASP, 29GLU
Model 4	−4.48 ± 0.57	23GLU, 27CYS, 39GLU, 40SER, 44GLY, 47TYR, 48ASN, 50THR	23GLU, 27CYS, 39GLU, 47TYR, 48ASN, 50THR
Model 5	−4.44 ± 0.74	23GLU, 24ARG, 26LEU, 28ASP, 39GLU, 41ILE, 48ASN, 50THR	23GLU, 24ARG, 26LEU, 28ASP, 39GLU, 41ILE, 48ASN, 50THR

## Data Availability

Not applicable.

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
