# Peer review of "Revealing Angiopep-2/LRP1 Molecular Interaction for Optimal Delivery to Glioblastoma (GBM)"

_molecules, 2022, doi:10.3390/molecules27196696_

Round 1

Reviewer 1 Report (New Reviewer)

The manuscript by Angela Costagliola di Polidoro et al., deals with the identification of Angiopep-2/LRP1 molecular interaction, aiming to clarify the impact of peptide conjugation on its transport mechanism. For this scope, molecular docking studies have been performed. Overall, the study is quite interesting but it must be improved in several points, especially in the technical part which is not well-organized. Therefore, the authors should address the following issues.

In the whole manuscript, please replace the term “amino acidic sequence” with the term “amino acid sequence”.

Introduction part

1. In the first paragraph of the introduction (lines 38-40) examples and references derived from the literature should be added.

2. Lines 53-55 should be supported by literature data. Please, add appropriate literature.

3. In the whole manuscript replace the abbreviation MD with Molecular Docking. According to the literature, the abbreviation MD is used for Molecular Dynamics simulations.

4. In line 103, the term “moiety” should be replaced by the term “site”.

Materials and Methods part

1. In lines 114-115, the links or the reference of the standard databases should be added.

2. The authors refer that the 3D model of ANG-2 is not available. However, in line 120, the constructed 3D model of the peptide is compared to experimental conformation. Could the authors give a further explanation.  

3. Also, could the authors complete some technical data? For example, the most probable conformations were subjected to energy minimization in order to produce a low energy conformation?  

4. In line 133, please remove “very”.

5. The PDB files of CR17 and CR56 were prepared, using a specific software? Please, give further information.

6. In the “Flexible docking by Autodock Vina” part of materials and methods, it is not clear for me if the authors used flexible or rigid docking? Could they give an explanation?   

7. The coordination  of gridbox must be completed.

Results part

1. In lines 212-213, the authors characterize the -5.68 kcal/mol binding energy as high. This value is low and not high. The authors must clarify the mentioned issue.

2. In line 250 (Figure 2), replace the term “binding site” with “binding mode”.

3. The title of Table 1 is not clear. Could you correct it? Also, the format of residues numbering is the opposite. For example Asp28.

4. The format of Figure 3 is different compared to the others. Also, in Model 4 the amino acids (axis x) are missed. Generally, the quality of the figures must be improved.

5. Generally, in my opinion the results must be improved in order to be more clear for the reader.

Discussion part and Conclusions

The discussion part lack of references. Please, complete with the appropriate references that confirm the discussion.

Author Response

Reviewer 2 Report (New Reviewer)

The paper deals with the modeling of the interaction between the synthetic polypeptide ANG-2 and the LRP-1 receptor present in the BBB endothelium. The topic is quite interesting as the targeting ability of the polypeptide is relevant in both healthy and damaged tissue due to the presence of glioblastoma. The use of methodologies based on Molecular Docking has given interesting results, which may have a general value as regards trafficking through the BBB. Therefore it is somewhat misleading to mention in the title the design of nanoparticles, which intends to recall some works of the research group, but which is not the focus of this work.

Therefore I suggest modifying the title, eliminating the design of the nanoparticles, which can be illustrated as an application example in the Conclusions, as well as eliminating it in the Discussion, in particular in lines 309, 325-326, and 350-351.

Lines 137-151 and 198-206, which describe LRP-1, should also be moved to the Introduction.

The sentence of row 157 and of row 196, which constitute the title of some paragraphs, must be written in italics.

In row 71 and 246 the author's name must be deleted.

Finally, the graph in Figure 1b is not legible.

Once these changes have been made, the document can be published.

Round 2

Reviewer 1 Report (New Reviewer)

Dear authors, 

the revised version of the manuscript have been addressed all recommended issues.

Therefore, I recommend the acceptance of the manuscript for publication.  

This manuscript is a resubmission of an earlier submission. The following is a list of the peer review reports and author responses from that submission.

Round 1

Reviewer 1 Report

The study is focus on a molecular docking (MD)-based strategy to investigate the  

molecular interaction of ANG-2 and the LRP-1 ligand-binding moieties (CR56 and CR 17), clarifying the impact of peptide conjugation on its transport mechanism. 

   The work is innovative. however the it is need to be major revised to be published in Biology

Q1 The authors said that they applied a molecular docking (MD)-based strategy to investigate the molecular interaction of ANG-2 and the LRP-1 ligand-binding moieties. However, molecular dynamics simulation was called MD, and  molecular docking was called docking!

Q2 The three-dimensional structure is obtained by the software PEP-FOLD3. How did the reliable of the structure? It need to be checked!

Q3 In page 5, line 162-163, they should add the reference for the PDB code.

Q4 Molecular dynamics simulations need to be used to explore the comformational changes between the ligands and the protein.

Reviewer 2 Report

The article performed molecular docking between two LRP-1 ligand binding moieties and five different version of ANG-2 peptide. The article is not novel enough and there are no major concussions from the study. Therefore, the article should not be published. My major concerns are-

1. In the docking studies, the authors showed that ANG-2 peptide interacts with the LRP-1 binding moieties which is already known by experimental studies as the authors indicated in the introduction. There are no in depth or further understanding regarding the mechanism in this study. 

2. This docking study will be highly dependent on the initial structure of the ANG-2 peptide. For example, model 1 and 3 are quite similar in their initial representation and they showed similar contact interaction with the LRP-1. How accurate this model representations are for the ANG-2 peptide needs to studied first.

3. Another major conclusion is that ANG-2 binds to the ASP929 of CR56. However, table 1 suggests that the peptide binds mostly with ASP residues. Also, which one is exactly the ASP929 here? I assume it is 28ASP as the model starts from SER932. In that case, model 2 and 3 did not particle in electrostatic interaction with CR56. So, this conclusion is not strongly evident by the results. 

If it is 29ASP, then the importance of 28ASP residue should also be taken into consideration since both the 28ASP and 29ASP participates in the PI-interaction for all the models. The authors could show the number of contacts between the ANG-2 residues and specifically for the these ASP residues and find out if they are significantly different.

4. Finally, there are no control experiments. For example, the authors should show some molecules that do not interact with LRP-1 moieties.

5. Also, there are some spelling mistakes in the introductions. The name CR56 is often mentioned as CR5.